# Recommendations for Diagnosis and Treatment of Lumbosacral Radicular Pain: A Systematic Review of Clinical Practice Guidelines

**DOI:** 10.3390/jcm10112482

**Published:** 2021-06-03

**Authors:** Ahmad Khoshal Khorami, Crystian B. Oliveira, Christopher G. Maher, Patrick J. E. Bindels, Gustavo C. Machado, Rafael Z. Pinto, Bart W. Koes, Alessandro Chiarotto

**Affiliations:** 1Department of General Practice, Erasmus MC, University Medical Center Rotterdam, 3000 CA Rotterdam, The Netherlands; a.khorami@erasmusmc.nl (A.K.K.); p.bindels@erasmusmc.nl (P.J.E.B.); b.koes@erasmusmc.nl (B.W.K.); 2Physical Therapy Department, Faculty of Medicine, University of Western São Paulo (UNOESTE), Presidente Prudente, Sao Paulo CEP 19060-900, Brazil; crystianboliveira@gmail.com; 3Institute for Musculoskeletal Health, Sydney Local Health District, Sydney, NSW 2050, Australia; christopher.maher@sydney.edu.au (C.G.M.); gustavo.machado@sydney.edu.au (G.C.M.); 4Sydney School of Public Health, Faculty of Medicine and Health, University of Sydney, Sydney, NSW 2050, Australia; 5Department of Physical Therapy, Universidade Federal de Minas Gerais (UFMG), Belo Horizonte 30000-000, Brazil; rafaelzambelli@gmail.com; 6Center for Muscle and Joint Health, University of Southern Denmark, 5230 Odense, Denmark

**Keywords:** lumbar radicular pain, clinical practice guidelines, AGREE II, diagnosis, treatment

## Abstract

The management of patients with lumbosacral radicular pain (LRP) is of primary importance to healthcare professionals. This study aimed to: identify international clinical practice guidelines on LRP, assess their methodological quality, and summarize their diagnostic and therapeutic recommendations. A systematic search was performed (August 2019) in MEDLINE, PEDro, National Guideline Clearinghouse, National Institute for Health and Clinical Excellence (NICE), New Zealand Guidelines Group (NZGG), International Guideline Library, Guideline central, and Google Scholar. Guidelines presenting recommendations on diagnosis and/or treatment of adult patients with LRP were included. Two independent reviewers selected eligible guidelines, evaluated quality with Appraisal of Guidelines Research & Evaluation (AGREE) II, and extracted recommendations. Recommendations were classified into ‘should do’, ‘could do’, ‘do not do’, or ‘uncertain’; their consistency was labelled as ‘consistent’, ‘common’, or ‘inconsistent’. Twenty-three guidelines of varying quality (AGREE II overall assessment ranging from 17% to 92%) were included. Consistent recommendations regarding diagnosis are (‘should do’): Straight leg raise (SLR) test, crossed SLR test, mapping pain distribution, gait assessment, congruence of signs and symptoms. Routine use of imaging is consistently not recommended. The following therapeutic options are consistently recommended (‘should do’): educational care, physical activity, discectomy under specific circumstances (e.g., failure of conservative treatment). Referral to a specialist is recommended when conservative therapy fails or when steppage gait is present. These recommendations provide a clear overview of the management options in patients with LRP.

## 1. Introduction

Low back pain (LBP) is globally not only a major medical problem but also a major economic problem [1]. Despite intensified research efforts on LBP management, the population burden and disability related to this disorder is increasing [2,3,4]. According to The Global Burden of Disease 2017 study, Years Lived with Disability due to LBP have globally increased by 54% between 1990 and 2015 [5]. LBP affects many people, especially female individuals and those aged 40–80 years, with a mean point prevalence of 11.9%, and 1-month prevalence of 23.2% [6]. Among patients with LBP seeking care in primary care, approximately 36% also report low back-related leg pain below the knee [7].

Low back-related leg pain is either radicular or referred (non-specific) pain. The former is described as radiating pain where a spinal nerve root is involved causing leg pain along the spinal nerve accompanied by numbness and tingling, muscle weakness and loss of reflexes. The latter is described as pain spreading down the legs arising from structures such as disc, joints or ligaments [8]. In the literature, multiple terms are used for lumbosacral radicular pain (LRP), with lumbar disc herniation being the most common diagnosis for this condition [9]. A large majority of patients with LRP tend to have a favourable prognosis in terms of pain and disability, but the time to recovery is usually longer than that of patients with LBP without concomitant radicular pain [7,10,11,12]. Therefore, an accurate assessment of these patients is needed to provide adequate management and treatment at an early stage of presentation.

Clinical practice guidelines are developed for implementing strong evidence into clinical care, to improve quality of care and reduce variation in decision making of healthcare practitioners. Over the last decades, an increasing number of guidelines have been developed in different countries for patients with LBP [13]. In most of these guidelines, diagnostic triage is recommended (i.e., classification into specific or non-specific LBP) [13], and some guidelines include diagnostic and therapeutic recommendations for different types of LBP. Additionally, an increasing number of guidelines containing specific recommendations for LRP have been issued over the past years.

Since 2001, overviews of clinical guidelines for the management of patients with LBP have been conducted and updated [13,14,15,16,17]. However, these overviews have focused on clinical recommendations for patients with acute or chronic non-specific LBP. No systematic review has been conducted on clinical practice guidelines for patients with LRP. Moreover, while the methodological quality of guidelines on non-specific LBP has been reviewed up to 2009 for acute LBP [14] and up to 2018 for chronic LBP [18], a quality assessment of guidelines focusing only on LRP has never been performed. Since 2009, the Appraisal of Guidelines Research & Evaluation (AGREE) II instrument can be used to assess the methodological quality of clinical practice guidelines [19].

The aim of this study was to retrieve all existing guidelines formulating recommendations on the clinical management of patients with LRP, to assess their methodological quality using the AGREE II tool, and to summarize their diagnostic and therapeutic recommendations.

## 2. Methods

### 2.1. Review Registration and Reporting

The protocol for the review was registered on the International Prospective Register of Systematic Reviews (PROSPERO) with ID number CRD42020138738. This review was reported in line with the Preferred Reporting Items for Systematic Reviews and Meta-Analyses (PRISMA) statement [20].

### 2.2. Literature Search

Literature searches were performed in the following electronic databases from their inception up to August 2019: MEDLINE via OVID, PEDro, National Guideline Clearinghouse, National Institute for Health and Clinical Excellence (NICE), New Zealand Guidelines Group (NZGG), International Guideline Library, and Guideline central. No language restrictions were applied. We also conducted web searches on Google Scholar using recommendations described elsewhere [21], but only the first 10 pages were screened because they retrieve the most relevant search results [21]. Furthermore, backward citation tracking on the reference lists of previous relevant reviews on the topic was performed [13,18]. Appendix B provides the search strategy used in each database.

### 2.3. Guidelines Selection

Clinical practice guidelines providing recommendations regarding diagnosis and treatment of LRP were included. Only guidelines formulating recommendations based on evidence were included. A guideline was included regardless of the type of professional association (e.g., physiotherapy, chiropractic, multidisciplinary), geographical location, and date of publication. The following terms were considered as LRP synonyms: sciatica, radiculopathy, nerve root compromise, nerve root compression, lumbar radicular syndrome, disc herniation, radiculitis, nerve root pain, and nerve root entrapment. Documents briefly mentioning LRP for diagnostic triage without providing further details on its management were excluded. When multiple versions of a guideline issued by a similar professional association were available, only the most recent version was selected. Clinical practice guidelines available in English, German, Portuguese, Spanish, Italian or Dutch were included because the author team could understand these languages. If guidelines in other languages were found, translators for extracting the required information from the guidelines were sought. If this method failed, documents for which no translators could be identified were excluded. Title/abstracts and full-texts were screened by two independent reviewers (AKK and CBO) and disagreements were discussed in online consensus meetings. If disagreements could not be solved, a third reviewer (AC) arbitrated.

### 2.4. Quality Assessment

The methodological quality of clinical practice guidelines was assessed using the AGREE II tool [22]. AGREE II is an update of the previous AGREE instrument to improve its measurement properties (i.e., reliability and validity), to refine its items and to improve the supporting documentation (i.e., original training manual and user’s guide) [19]. AGREE II consists of 23 items categorized in six domains: scope and purpose; stakeholder involvement; rigor of development; clarity of presentation; applicability; editorial independence. Each item is scored on a 7-point Likert scale from one (i.e., strongly disagree) to seven (i.e., strongly agree). AGREE II also includes two global items. The first global item is scored similarly to the 23 items and evaluates a guideline overall quality. The second item evaluates the recommendation for use which is rated using a three-point scale (i.e., yes, yes with modification, or no). A previous study [23] using four appraisers found good to excellent inter-rater reliability (intraclass correlation coefficient [ICC_2,1_] ranging from 0.66 to 0.93) for the 23 items and the first global item. The AGREE II manual recommends the appraisal of the guidelines by at least two reviewers [19]. Thus, two independent authors (AKK and CBO) performed the online AGREE II training [24] and followed the AGREE II manual to assess each included guideline [19]. To calculate the score for each domain all the scores of the individual items in a domain from both appraisers were summed up and scaled as a percentage of the maximum possible score. The overall assessment score is the mean score for the 6 different domains. To investigate the reliability among the assessors, we calculated inter-rater reliability from the two appraisers of the scores obtained for each domain and first global item using the ICC_2,1_ and 95% confidence interval. Inter-rater agreement was classified as; 0.75–1.00, excellent; 0.60–0.74, good; 0.40–0.59, fair and <0.40 as poor [25]. Reliability analyses were performed using IBM SPSS Statistics 25 with a two-way random effects model for the domain scores and the first global item scores.

There is not consensus on categorizing guidelines as high, average or low quality guidelines depending on AGREE II scores. However, several methods are provided in the AGREE II manual. A previously described threshold of ≥60% for an acceptably high score on a domain [26] was adopted in this review; for the overall quality of the included guidelines: high-quality guideline when ≥5 domains were scored ≥60%, average quality guideline when 3 or 4 domains were scored ≥ 60% and low-quality guideline when ≤2 domains scored ≥ 60%.

### 2.5. Data Extraction and Synthesis

Two authors (AKK and CBO) independently extracted data from the included guidelines. In cases of no consensus, a third reviewer (AC) was consulted. The following information was extracted using a standardized form: recommendations regarding diagnosis (e.g., history, physical examination) and treatment (e.g., patient education, pharmacological intervention). Regarding surgical treatment options only recommendations for discectomy were extracted as this procedure is the most common applied for disc herniation. As radicular pain is also covered in most generic LBP guidelines, recommendations were extracted when it was clearly specified that they concerned radicular pain, or if recommendations for LBP in general also applied to radicular pain. If available, the level of evidence considered to formulate each recommendation was also extracted.

In order to assess the type and direction of the recommendations, the extracted recommendations were firstly classified into one of the categories: ‘Should do’, ‘Could do’, ‘Do not do’ and ‘Uncertain’. This classification was dependent on the terminology used for a recommendation in the guideline (see Appendix C, Table A1). Secondly, to determine the consistency of a recommendation the following categories were identified using a modified version of the approach previously adopted by Lin et al. [27]:a.Consistent recommendations: from the guidelines including recommendation for a specific approach, the majority (≥80%) indicate as ‘should do’, ‘could do’, ‘do not do’, or ‘uncertain’, but without conflicting recommendations across guidelines. Conflicting recommendations are present when at least one ‘should do’ or ‘could do’, and at least one ‘do not do’ is applied for the same recommendation in different guidelines.b.Common recommendations: from the guidelines including recommendation for a specific approach, most (between 50% and 80%) indicate as ‘should do’, ‘could do’, ‘do not do’, or ‘uncertain’, but with no conflicting recommendations across guidelines.c.Inconsistent recommendations: a recommendation for one approach indicates ‘should do’ or ‘could do’, and another recommendation for the same approach indicates ‘do not do’ or ‘uncertain’, both recommendations issued by different guidelines; the same applies if a recommendation for an approach is ‘uncertain’, and another recommendation for the same approach is ‘do not do’.

To determine the consistency across recommendations, only the options included at least two different guidelines were considered.

## 3. Results

### 3.1. Guidelines Selection

Our literature searches identified 3032 records. After screening of title/abstract and full texts, 23 eligible guidelines were included (Figure 1). The characteristics of these guidelines are presented in Table 1. The 23 included guidelines [28,29,30,31,32,33,34,35,36,37,38,39,40,41,42,43,44,45,46,47,48,49,50] were developed in 10 different countries from 3 continents (i.e., North America, Europe and Asia): United States (*n* = 12, 52%), Canada (*n* = 2), Belgium (*n* = 1), Denmark (*n* = 1), Korea (*n* = 1), Philippines (*n* = 1), Italy (*n* = 1), the Netherlands (*n* = 1) and Norway (*n* = 1). One guideline was a joint European guideline. One guideline was written in Dutch [45], one updated guideline in Norwegian [44] and the other 21 in English. The professional entities involved in developing the guidelines vary in different countries (Table 1). Most guidelines (*n* = 14, 61%) are from a specific (medical) professional association (e.g., general practitioners, pain physicians, radiologists, chiropractors, physiotherapists).

### 3.2. Quality Assessment and Inter-Rater Agreement

In Table 2, the AGREE II scores for each domain and the overall assessment scores are displayed for each included guideline. The overall quality score was variable, ranging from 17% to 92%, with the NICE guideline having the highest score and the Department of Labor and Employment, Division of Worker’s Compensation (DLW-DWC)the lowest score. Of the 23 included guidelines, 15 (65%) scored above the ≥ 60% threshold in the overall assessment score. According to our described classification criteria [26], ten [32,34,38,40,42,43,44,46,48,50] (43%) of the guidelines are classified as high quality, seven [29,30,33,35,37,39,41] (30%) as average quality, and six [28,31,36,45,47,49] (26%) as low quality. Five guidelines (American Pain Society (APS) and Institute for Clinical Systems Improvement (ICSI) from USA, NICE from UK, Belgian Health Care Knowledge Center (KCE) from Belgium, and Institute of Health Economics Toward Optimized Practice (TOP) from Canada) displayed a score ≥ 60% in all 6 domains (Table 2).

The domains with the highest scores were “Clarity of presentation” and “Scope and purpose” with mean scores of 85% and 77%, respectively (Table 2). The domains that were less well addressed by guideline developers were “Applicability” and “Stakeholder involvement” with mean scores of 36% and 55%, respectively. “Editorial independence” is the domain with the highest variability among the guidelines ranging from 0% to 100% (Table 2).

Table 3 indicates inter-rater agreement for AGREE II domains and overall scores. The ICC_2,1_ was ‘excellent’ for the domains “Scope and purpose”, “Stakeholder involvement”, “Rigor of development”, “Applicability”, “Editorial independence” and “Overall rating”, and ‘fair’ for the domain “Clarity of presentation”. Due to the small sample of assessed guidelines (*n* = 23), ICC 95% confidence intervals were broad, especially for the domain with the lowest inter-rater agreement (Table 3).

### 3.3. Recommendations for Diagnosis

Table 4 and supplementary Appendix A describe the recommendations for physical examination and other diagnostic procedures in each clinical practice guideline.

#### 3.3.1. Physical Examination

A minority of guidelines (6 out of 23; 26%); [30,39,42,44,45,47] made recommendations concerning physical examination (Table 4). The consistent recommendation for ‘should do’ in the physical examination are: performing straight leg raise test (SLR) [30,39,42,44,45,47], crossed SLR test [39,42,44,45,47], mapping pain distribution [39,47], steppage gait (inability to lift the foot while walking due to the weakness of muscles that cause dorsiflexion of the ankle joint) assessment [39,47], and agreement of signs and symptoms [39,47].

For the recommendation ‘muscle testing’, the guidelines are evenly distributed with ‘should do’ and ‘could do’ [30,39,42,44,45,47]. The inconsistent recommendations concerned the performance of: femoral stretch test [42,44,47], reflex tests [30,38,42,44,47] and slump test [42,47].

#### 3.3.2. Diagnostics

A majority of the guidelines (19 out of 23, 82%; [28,29,30,31,35,36,37,38,39,40,42,43,44,45,46,47,48,49,50]) made recommendations concerning diagnostics (Table 4 and supplementary Appendix A). A consistent recommendation for ‘should do’ in imaging is to perform a computed tomography (CT) scan when history and physical examination findings are consistent with disc herniation, after 4–6 weeks of pain, if surgery is considered or severe or progressive neurologic signs and symptoms are present [28,30,31,39,42,44,47,50]. A common recommendation for ‘should do’ concerns magnetic resonance imaging (MRI) when history and physical examination findings are consistent with disc herniation, radiculopathy persists after six weeks, if surgery is considered, severe or progressive neurologic signs and symptoms are present or where an epidural glucocorticosteroid injection is being considered [28,29,30,31,37,39,42,43,44,47,48,49,50].

Consistent recommendations for ‘do not do’ are: routinely offering imaging in primary care or in the absence of red flags [28,31,35,38,40,44,46] and routine computed tomography/magnetic resonance imaging (CT/MRI) scans in the first 4–6 weeks [28,38,39,44,45,47]. There were inconsistent recommendations regarding the use of electromyography (EMG) [28,37,39,47,49], sensory nerve somatosensory evoked potentials (SEP) [36,42], discography [28,36] and diagnostic medial branch block [28,32].

### 3.4. Guideline Recommendations for Treatment

Table 5 and supplementary Appendix A describes the three treatment categories (i.e., non-invasive, pharmacological and invasive) provided by each clinical practice guideline.

#### 3.4.1. Non-Invasive Treatments

Most guidelines (16 out of 23; 70%; [28,30,33,35,36,38,39,40,41,42,43,44,45,46,47,49]) included recommendations regarding non-invasive treatments. The consistent recommendation for ‘should do’ is: educational care [30,38,44,45,46] and the common recommendation for ‘should do’ is physical activity [30,39,43,44,45,46,47,49]. Exercise/physical therapies [28,33,35,40,43,44] is consistently recommended as ‘could do’. The consistent recommendations for ‘do not do’ are: devices (e.g., belts, corset, foot orthotics etc.) [28,40,44,46] and transcutaneous electrical nerve stimulation (TENS)/percutaneous electrical nerve stimulation (PENS)/interferential therapy [28,39,40,44,46,47]. The inconsistent recommendations are: bed rest [28,39,44,45,47,49], acupuncture [28,38,39,41,43,44,46,47], traction [28,33,36,38,42,44,46,47], manipulations/mobilisations/soft tissue techniques [28,33,35,36,38,39,40,42,43,44,45,46,47], massage [28,39,44,47], therapeutic ultrasound [40,46,47] and heat/cold/infrared therapies [28,38,47,49].

#### 3.4.2. Pharmacological Interventions

Nine guidelines (out of 23; 39%; [28,38,39,40,44,45,46,47,49]) included recommendations regarding pharmacological interventions. The only consistent recommendation is for ‘do not do’ for medicinal cannabis [45,46]. For all the other medications, such as paracetamol, Non-steroidal anti-inflammatory drugs (NSAIDs), opioids, anticonvulsants, muscle relaxants, antidepressants, corticosteroids and antibiotics, the recommendations are highly inconsistent. For example, while six guidelines [28,38,39,40,45,47] tend to suggest NSAIDs as an option, one guideline [44] found no evidence to make a recommendation. Six guidelines [28,39,44,45,47,49] suggest paracetamol, but KCE [40] advices against it. NICE [46] advices against the use of strong opioids such as morphine or long term use of Tramadol in a non-specializing setting, but the Dutch General Practitioners Society (NHG) [45] guideline suggests morphine or fentanyl for chronic pain. Four guidelines [39,44,46,47] recommend a weak opioid such as tramadol as an option, where KCE [40] advises against any opioid.

#### 3.4.3. Invasive Treatments and Referral

Thirteen guidelines (out of 23; 57%; [28,32,36,38,39,40,42,43,44,45,46,47,50]) included recommendations regarding invasive treatments and three guidelines (out of 23; 13%; [39,45,47]) regarding referral. The common recommendation in invasive therapy for ‘could do’ is discectomy when conservative therapy fails or when progressive/persistent disability is present [28,32,36,39,42,43,44,45]. The recommendation regarding epidural injections is inconsistent across the guidelines [28,29,32,38,40,42,43,44,45,46,47,50]. The consistent recommendation for ‘should do’ is referral to a specialist when there is no improvement of symptoms with conservative therapy, or immediately when there is steppage gait [39,45,47].

## 4. Discussion

This systematic review provides a summary of the diagnostic and therapeutic recommendations from 23 international clinical practice guidelines for LRP. The consistent and common recommendations for ‘should do’ for physical examination are performing the SLR test, the crossed SLR test, mapping pain distribution, steppage gait assessment, and evaluating congruence of signs and symptoms. Regarding imaging, guidelines recommend CT scan or MRI under specific circumstances (e.g., physical examination findings are consistent with disc herniation, after 4–6 weeks of pain, surgery or epidural injections are considered, severe or progressive neurologic signs and symptoms present), and do not recommend the routine use of any form of imaging. The consistent and common therapeutic recommendations for ‘should do’ are: providing educational care and physical activity, referral to a specialist when conservative therapy fails or when steppage gait is present.

This systematic review provides a methodological quality assessment of the 23 selected guidelines using the AGREE II tool. The overall quality of the guidelines ranged from low to high. High quality guidelines (5 out of 9) are from (national) health care institutes [38,40,43,46,48], two from a pain society [32,34], one from a department of veterans affair [50], one from interdisciplinary back pain network [44] and one from the spine society [42]. In this study, “Clarity of presentation” and “Scope and purpose” were the AGREE domains with the highest scores, and “Applicability” and “Stakeholder involvement” with the lowest scores. These finding are in accordance with another critical appraisal of the quality of LBP guidelines [51]. This previous study assessed 5 guidelines [33,38,46,48,49] which were also included in this review (although we included the latest version of the ICSI [38] guideline from 2018). Compared to the previous review, four out of five of these guidelines were classified as the same quality (i.e., high). One guideline [49] was classified as low quality in this review but of average quality in the other. This discrepancy could be due to a difference in the amount of AGREE II appraisers which could lead to a higher ICC ratio (two appraisers in this study vs. 4 appraisers in Doniselli et al. [51]. Four guidelines [33,46,48,49] included in this review were also included in another review of LBP guidelines [18], where the domain scores were generally similar for the most of the domains. If we apply the same threshold for the domain scores, these 4 guidelines would be categorized with the same quality as in our study. Based on the AGREE II scores of this and earlier studies, it is important that guideline developers take potential barriers of implication of recommendations into account and provide criteria for monitoring and auditing (i.e., AGREE II applicability). Additionally, it is important to include individuals from all relevant professions and take the view of the target population into account (i.e., AGREE II stakeholder involvement).

Although many organizations tend to develop a new guideline, studies have been undertaken on adopting and adapting existed good quality guidelines for saving time and other resources. Schünemann H. et al. [52] has developed the “GRADE-ADOLOPMENT” approach for adopting, adapting and de novo development of recommendations for guideline productions. This approach allows guideline developers to quickly and efficiently create recommendations appropriate for their context where the evidence is taken into account. The Belgian guideline KCE is a good example where the UK comprehensive guideline (NICE) has been adopted and adapted for the Belgian population. Harstall C. et al. [53] have described a multidisciplinary adaption process for creating a single overarching evidence-based clinical practice guideline for patients with LBP. The adolopment approach could facilitate other national and international guideline developers to save time and other important resources and we suggest this approach for LBP when resources are limited and good quality guidelines already exists.

The guidelines commonly recommend physical examination such as performing a SLR test, crossed SLR test, muscle and sensory testing, and reflex test for lumbar radicular pain (Table 4). However, the sensitivity and specificity of these tests have been questioned. A systematic review [54] showed poor diagnostic performance of most physical tests to identify disc herniation when used in isolation, while better performance could be obtained when tests are combined. Two stretch tests that have shown high diagnostic accuracy in patients with LRP are the SLR test and the slump test [55] where the later test was found to be more sensitive. However, according to our finding the recommendation regarding the slump test is inconsistent and six guidelines suggested the SLR test (Table 4). Professionals involved in developing or updating guidelines should consider also the slump test for recommendation. The guidelines commonly/consistently recommend physical activity and exercises/physical therapy. These recommendations were also taken up by a recent narrative review [56] where it is concluded that both physical activity and structured exercises might be beneficial elements in conservative management of patients with lumbar radicular pain.

Besides the consistent and common recommendations for diagnostic and therapeutic options for LRP, we have also identified inconsistent recommendations among the guidelines. Regarding non-invasive treatments, there are inconsistent recommendations on advising bed rest, acupuncture, traction, manipulations/mobilisations/soft tissue techniques, massage, therapeutic ultrasound and heat/cold therapy. Three guidelines ([28,39,47], two low quality and one average quality) do not recommend bed rest with the exception of 2–4 days in severe cases, while two other guidelines ([44,45], one high and one low quality) recommend bed rest for a few days to relieve pain. However, other high quality guidelines [32,34,38,40,42,43,46,48,50] do not include any recommendations on bed rest. Therefore, the question could be raised how strong bed rest recommendation is as a non-invasive therapeutic option. For pharmacological interventions, there are inconsistent recommendations for paracetamol, NSAIDs, opioids, anticonvulsants, muscle relaxants, antidepressants, corticosteroids and antibiotics (Table 5). These inconsistencies in guideline recommendations are not surprising considering that there is still uncertain evidence regarding their effectiveness for patients with LRP [57]. Inconsistent recommendations for the use of paracetamol is not surprising. In fact, the publication of the placebo controlled trial of paracetamol (PACE) study from 2014, probably explain these differences [58], as no effect was detected favoring paracetamol on pain and speed recovery in patients with acute LBP with or without leg pain. It can be noticed that guidelines published earlier than 2014 recommend paracetamol, whereas recently published guidelines are being careful with making this recommendation. Only the NHG guideline from 2015 [45] still suggests paracetamol as the first analgesic choice. This could lead to a conclusion that most guidelines might have followed the PACE study results considering that a randomized controlled study in patients with LRP is missing. Nevertheless, such study should be conducted as it would provide clearer indications on the efficacy of this drug for future LRP guidelines. A Cochrane review from 2016 [59] showed no significant efficacy of NSAIDs for pain reduction in treating patients with LRP. However, two guidelines [38,40] issued after 2016 suggest NSAIDs to be considered. More research is needed to evaluate the effects of NSAIDs on this condition to reach more firm conclusions. In the category invasive treatments, the recommendation for epidural injections is inconsistent. A recently published Cochrane review [60] concluded that there are small effects of epidural injections in the treatment of lumbar radicular pain, which are mainly evident at short-term follow up.

To the best of our knowledge, this is the first systematic review of clinical practice guidelines focusing on diagnosis and treatment of LRP. A strength of this study is using the AGREE II tool to methodologically assess the quality of the included guidelines and consistent findings in AGREE with other two recent reviews [18,51]. We have transparently reported all recommendations in every single guideline in the Tables (Appendix A). A limitation of this review is that it was not possible to assess the eligibility of three guidelines due to the review team not being able to read and understand the language of these guidelines (one Hebrew and two Croatian). Another limitation is that AGREE II appraisal of the NHG and the Norwegian Back Pain Network (NBPN) guidelines was done by only one appraiser because one of the two reviewers could not understand the Dutch or Norwegian language. In assessing the consistency and clinical inference of the recommendation we have not taken the publication date of a guideline into account. This approach could raise the consideration that this could lead to potentially biased recommendations as an older guideline is equally weighted as a recently published guideline based on more recent evidence. Nevertheless, we performed a sensitivity analysis in a sample of five recommendations from Table 4 and Table 5 (i.e., straight leg test, performing CT scan, bed rest, physical activity and paracetamol) in which guidelines published before 2010 were excluded. The consistency of recommendations and clinical inference remained the same. Therefore, the publication date of guideline would not influence the results consistently. We acknowledge the limitation of the AGREE II consortium not setting specific cut-off scores for domains or guidelines of high versus low quality. The cut-off scores used in this review were taken from a small study [26], but they were also used in other recent reviews of clinical practice guidelines using the AGREE II [61,62,63]. More research on the optimal cut-off scores for the domain and total scores is needed.

This review has highlighted the lack of homogeneity in the manner in which clinical practice guidelines formulate the strength of their recommendations, and the related level of evidence per recommendation. This makes it difficult, at times, to compare and contrast the strength of each recommendation from different guidelines. In our study, we have defined our own terminology by grouping different terms used by the different guidelines (Appendix C). In our view, this point could be addressed by issuing a standardized terminology for strength of a recommendation when developing a clinical practice guideline. Moreover, to date, no standardized threshold value has been suggested to classify high- and low-quality guidelines using the AGREE II appraisal. Introducing a standardized quality classification system based on the AGREE II domain scores is a point of consideration for the future.

## 5. Conclusions

Twenty-three clinical practice guidelines for patients with LRP were retrieved and their overall quality ranged from low to high according to the AGREE II tool. These guidelines recommend physical examinations perform the SLR test, crossed SLR test, mapping pain distribution, steppage gait assessment, and agreement of signs and symptoms. Imaging is only recommended under specific circumstances, and its routine use is consistently not recommended. For treatment, the recommendations are: providing educational care, prescribing physical activity, and referring to a specialist when conservative therapy fails, or when steppage gait is present. These consistent recommendations should be adopted by healthcare professionals and healthcare systems worldwide to implement the most effective care.

## Figures and Tables

**Figure 1 jcm-10-02482-f001:**
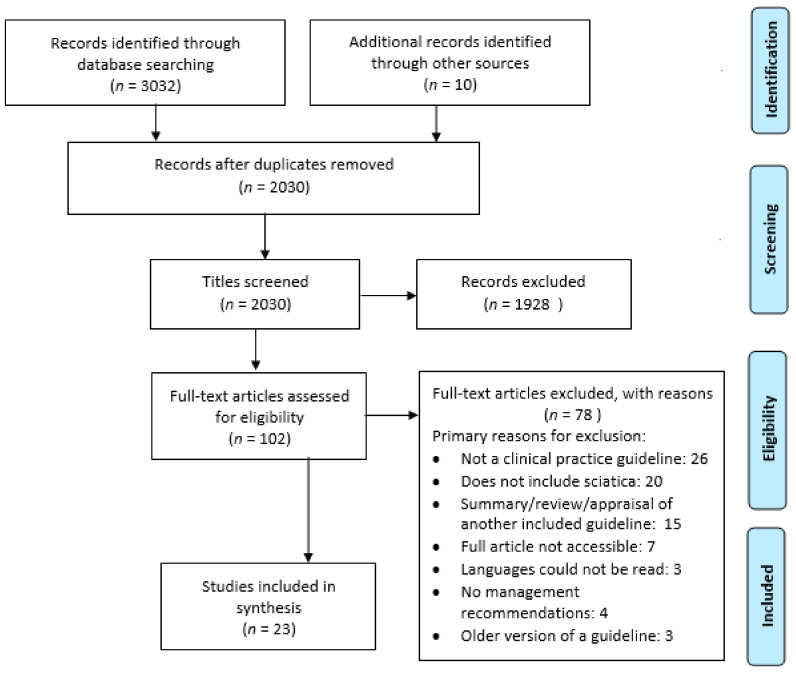
Preferred Reporting Items for Systematic Reviews and Meta-Analyses (PRISMA) flow diagram.

**Table 1 jcm-10-02482-t001:** Characteristics of included clinical practice guidelines on recommendations for patients with lumbar radicular pain (*n* = 23).

Title Guideline	Year	Country	Professional Bodies/ Abbreviation
Low back disorders	2016	USA	American College of Occupational and Environmental Medicine (ACOEM [28])
Diagnostic Imaging for Low Back Pain: Advice for High-Value Health Care From the American College of Physicians	2011	USA	American College of Physicians (ACP [29])
Diagnosis and Treatment of Low Back Pain: A Joint Clinical Practice Guideline from the American College of Physicians and the American Pain Society	2007	USA	American College of Physicians and the American Pain Society (ACP-APS [30])
ACR Appropriateness Criteria Low Back Pain	2016	USA	American College of Radiology (ACR [31])
Interventional Therapies, Surgery, and Interdisciplinary Rehabilitation for Low Back Pain	2009	USA	American Pain Society (APS [32])
Low Back Pain: Clinical Practice Guidelines Linked to the International Classification of Functioning, Disability, and Health from the Orthopedic Section of the American Physical Therapy Association	2012	USA	Orthopedic Section of the American Physical Therapy Association (APTA [33])
An Update of Comprehensive Evidence-Based Guidelines for Interventional Techniques in Chronic Spinal Pain. Part II: Guidance and Recommendations	2013	USA	American Society of Interventional Pain Physicians (ASIPP [34])
Spinal Manipulative Therapy and Other Conservative Treatments for Low Back Pain: A Guideline From the Canadian Chiropractic Guideline Initiative	2018	Canada	Canadian Chiropractic Guideline Initiative (CCGI [35])
Low Back Pain medical Treatment Guidelines	2014	USA	Department of Labor and Employment, Division of Worker’s Compensation (DLE-DWC [36])
European guidelines for the management of chronic non-specific low back pain	2004	Europe	European Commission, Research Directorate-General, department of Policy, Coordination and Strategy (EG [37])
Adult Acute and Subacute Low Back Pain Diagnosis Algorithm	2018	USA	Institute for Clinical Systems Improvement (ICSI [38])
Diagnostic therapeutic flow-charts for low back pain patients: the Italian clinical guideline	2006	Italy	Italian Health Ministry-Care and Research Institute Fondazione Don Carlo Gnocchi ONLUS of Milan (IHM [39])
Low back pain and radicular pain: assessment and management	2017	Belgian	Belgian Health Care Knowledge Center (KCE [40])
Korean medicine clinical practice guideline for lumbar herniated intervertebral disc in adults	2017	Korea	The Korea Institute of Oriental Medicine (KIOM [41])
Clinical Guideline for the Diagnosis and Treatment of Lumbar Disc Herniation with Radiculopathy	2012	USA	North American Spine Society (NASS [42])
National Clinical Guideline: interventions for recent onset lumbar radiculopathy	2016	Denmark	National Board of Health (Denmark) (NBHD [43])
Acute low back pain Interdisciplinary clinical guidelines	2007	Norway	The Norwegian Back Pain Network and an interdisciplinary working group (NBPN [44])
NHG-Standaard Lumbosacraal radiculair syndroom	2015	Netherlands	Dutch General Practitioners Society (NHG [45])
Low back pain and sciatica in over 16s: assessment and management	2016	UK	National Institute for Health and Care Excellence (NICE [46])
Clinical Practice Guidelines on the Diagnosis and Management of LBP	2011	Philippines	Philippine Academy of Rehabilitation Medicine PARM [47])
Evidence-Informed Primary Care Management of Low Back Pain	2015	Canada	Institute of Health Economics Toward Optimized Practice (TOP [48])
Acute Low Back Pain	2010	USA	The University of Michigan Health System (UMHS [49])
VA/Do clinical practice guideline for diagnosis and treatment of low back pain	2017	USA	Department of Veterans Affairs and Department of Defense (Va/Dod [50])

**Table 2 jcm-10-02482-t002:** Appraisal of Guidelines Research & Evaluation (AGREE ) II domain scores and quality of the eligible guidelines (*n* = 23).

Guideline			AGREE Domains					
Guideline *	Scope and Purpose, %	Stakeholder Involvement, %	Rigor of Development, %	Clarity of Presentation, %	Applicability, %	Editorial Independence, %	Overall Assessment, %	Quality
ACOEM	58	44	50	100	23	92	61	Low
ACP	86	36	11	83	46	100	60	Average
ACP-APS	97	47	76	94	46	96	76	Average
ACR	14	17	81	67	0	17	33	Low
APS	100	67	94	94	60	96	85	High
APTA	81	47	72	89	0	0	48	Average
ASIPP	83	64	72	67	29	96	68	High
CCGI	92	72	74	58	33	67	66	Average
DLE-DWC	17	11	0	72	2	0	17	Low
EG	72	64	70	92	33	38	61	Average
ICSI	83	69	76	92	60	92	79	High
IHM	61	58	68	83	50	25	58	Average
KCE	97	75	88	86	77	100	87	High
KIOM	78	28	76	81	6	83	59	Average
NASS	83	61	75	78	17	92	68	High
NBHD	86	47	91	97	60	96	80	High
NBPN	89	83	83	94	29	100	80	High
NHG	83	56	58	89	25	33	57	Low
NICE	97	94	95	97	71	100	92	High
PRAM	72	25	31	81	38	0	41	Low
TOP	94	78	84	94	73	96	87	High
UMHS	56	25	57	94	6	92	55	Low
VA/Dod	94	92	77	83	44	67	76	High
Mean	77	55	68	85	36	69	65	

* Table 1 provides the extensive names of the included guidelines.

**Table 3 jcm-10-02482-t003:** Inter-rater agreement for AGREE II domains and overall rating.

Domain	ICC * (95% CI)
Scope and purpose	0.847 (0.631 to 0.936)
Stakeholder involvement	0.820 (0.2563 to 0.926)
Rigor of development	0.858 (0.636 to 0.943)
Clarity of presentation	0.549 (-0.044 to 0.811)
Applicability	0.874 (0.695 to 0.948)
Editorial independence	0.901 (0.762 to 0.959)
Overall rating	0.785 (0.493 to 0.910)

* ICC; Intraclass Correlation Coefficient.

**Table 4 jcm-10-02482-t004:** Guideline recommendations for physical examination and other diagnostic procedures.

Physical Examination	Guideline *	Consistency	Clinical Inference
Femoral stretch test	NASS	Inconsistent	None
	NBPN		
	PARM		
Straight leg test	NASS	Consistent	Should do
	PARM		
	IHM		
	NHG		
	ACP-APS		
	NBPN		
Crossed straight leg test	NASS	Consistent	Should do
	PARM		
	IHM		
	NHG		
	NBPN		
Muscle testing	NASS	Common/consistent	Should/Could do
	PARM		
	IHM		
	NHG		
	ACP-APS		
	NBPN		
Sensory testing	NASS	Common	Could do
	PARM		
	IHM		
	ACP-APS		
	NBPN		
Reflex tests (ankle and knee tendon)	PARM	Inconsistent	None
	NBPN		
	ACP-APS		
	NASS		
Mapping pain distribution	PARM	Consistent	Should do
	IHM		
Slump test	PARM	Inconsistent	None
	NASS		
Wasserman test	PARM	Consistent	Could do
	IHM		
Gait	PARM	Consistent	Should do
	IHM		
Agreement of signs and symptoms	PARM	Consistent	Should do
	IHM		
Diagnostics			
Imaging			
*Routinely offering imaging in primary care or absent of red flags*	KCE ICSI NICE ACOEM CCGI ACR NBPN	Consistent	Do not do
*Computed Tomography (CT)/ Magnetic resonance imaging (MRI) routinely in first 4–6 weeks*	ICSI	Consistent	Do not do
	ACOEM		
	NHG		
	PRSM		
	IHM NBPN		
*CT when history and physical examination findings consistent with disc herniation, after 4–6 weeks of low back pain if surgery is considered, severe or progressive neurologic signs and symptoms present*	NASS VA/Dod TOP PRAM ACOEM IHM ACP-APS NBPN ACR	Consistent	Should do
*MRI when history and physical examination findings consistent with disc herniation, radiculopathy persists after six weeks, if surgery is considered, severe or progressive neurologic signs and symptoms present, where an epidural glucocorticosteroid injection is being considered*	NASS VA/DoD TOP NBHD EG PRAM ACOEM IHM ACP-APS UMHS NBPN ACR ACP	Common	Should do
Others			
*EMG*	EG PRAM ACOEM IHM UMHS	Inconsistent	None
*Sensory nerve somatosensory evoked potentials (SEP)*	NASS DLE-DWC	Inconsistent	None
*Discography*	ACOEM	Inconsistent	None
	DLE-DWC		
*Diagnostic medial branch block*	APS	Inconsistent	None
	ACOEM		

* Table 1 provides the extensive names of the included guidelines.

**Table 5 jcm-10-02482-t005:** Therapeutic recommendations from guidelines for lumbar radicular pain.

Non-Invasive Interventions	Guideline *	Consistency	Clinical Inference
Bed rest	ACOEM	Inconsistent	None
	PARM		
	IHM		
	NHG		
	NBPN		
Physical activity	NBHD	Common	Should do
	PARM		
	IHM		
	ACP-APS		
	NICE		
	NHG		
	NBPN		
Educational care	NICE	Consistent	Should do
	ICSI		
	NHG		
	ACP-APS NBPN		
Multidisciplinary approach/rehabilitation program/Psychological therapy	NICE	Common	Could do
	KCE		
	UMHS		
	NBPN		
Alternative medicine			
*Acupuncture*	ICSI	Inconsistent	None
	PARM		
	ACOEM		
	IHM		
	NICE		
	KIOM NBPN		
Manual therapies			
*Traction*	NASS	Inconsistent	None
	KCE		
	PRAM		
	ACOEM		
	DLE-DWC		
	NICE		
	NBPN		
	APTA		
*Manipulation/mobilisation/soft-tissue techniques*	NASS	Inconsistent	None
	KCE		
	ICSI		
	NBHD		
	PARM		
	ACOEM		
	DLE-DWC		
	IHM		
	NICE		
	NHG		
	CCGI		
	NBPN		
	APTA		
*Massage*	PRAM	Inconsistent	None
	ACOEM		
	IHM NBPN		
Devices	NICE	Consistent	Do not do
(e.g., belts, corset, foot orthotics etc.)	KCE		
	ACOEM		
	NBPN		
Exercise/physical therapies	NASS	Consistent	Could do
	KCE		
	ACOEM		
	NBHD		
	CCGI		
	NBPN		
	APTA		
Electrotherapies			
*TENS/PENS/interferential therapy*	NICE	Consistent	Do not do
	KCE		
	PRAM		
	ACOEM		
	IHM NBPN		
Therapeutic ultrasound	KCE	Inconsistent	None
	PARM		
	NICE		
Heat/cold/infrared therapies	ICSI	Inconsistent	None
	PRAM		
	UMHS		
	ACOEM		
**Pharmacological interventions**			
Paracetamol	KCE	Inconsistent	None
	PARM		
	ACOEM		
	IHM		
	UMHS		
	NHG		
	NBPN		
Non-steroidal anti-inflammatory drugs (NSAIDs)	KCE	Inconsistent	None
	ICSI		
	PARM		
	ACOEM		
	IHM		
	NHG		
	NBPN		
Opioids	NICE	Inconsistent	None
	KCE		
	PARM		
	IHM		
	NHG NBPN		
Paracetamol + opioids	IHM	Inconsistent	None
	NHG		
	NBPN		
Anti-epilepticum	KCE	Inconsistent	None
	VA/Dod		
	PARM		
	ACOEM		
	NASS		
	NICE		
	NHG		
Muscle relaxants	KCE	Inconsistent	None
	ICSI		
	PARM		
	ACOEM		
	IHM		
	NHG		
	NBPN		
Antidepressants	NASS	Inconsistent	None
	NICE		
	KCE		
	ACOEM		
	NHG		
Corticosteroids	VA/Dod	Inconsistent	None
	PARM		
	ACOEM		
	DLE-DWC		
	NHG		
	ACP-APS		
Antibiotics	KCE	Inconsistent	None
	ACOEM		
Cannabis	NICE	Consistent	Do not do
	NHG		
**Invasive Treatments**			
Surgery	NASS	Common	Could do
	APS		
	NHG		
	DLE-DWC		
	IHM		
	ACOEM NBPN		
Injection therapies			
*Epidural injections*	NICE	Inconsistent	None
	NASS		
	KCE		
	VA/Dod		
	ICSI		
	NBHD		
	PRAM		
	ACOEM		
	ACP		
	NHG		
	NBPN		
	APS		
Referral	PRAM	Consistent	Should do
	IHM		
	NHG		

* Table 1 provides the extensive names of the included guidelines.

## Data Availability

Data sharing is not applicable to this article.

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
