# Peer review of "Recommendations for Diagnosis and Treatment of Lumbosacral Radicular Pain: A Systematic Review of Clinical Practice Guidelines"

_jcm, 2021, doi:10.3390/jcm10112482_

Round 1

Reviewer 1 Report

Dear Authors, 

Thank you for your putting in those edits - looks excellent overall. Because I picked this before, so couldn't let that slip this time as well - line 359 still contains a typo "message" (instead of "massage"). Good luck

Author Response

  Thank you for your positive comments.  

Reviewer's comment: Because I picked this before, so couldn't let that slip this time as well - line 359 still contains a typo "message" (instead of "massage"). Good luck

Authors' response: Amended (line 362).      

Reviewer 2 Report

This study suggested very interesting results and experimental suggestions. 
Also, this paper is well written with logical flow. 
This paper deserves to be published.

Author Response

Thank you for your very positive view of our manuscript.

Reviewer 3 Report

I think the paper has improved substantially. Some minor issues remain:

As I previously commented, radicular pain spreads along the spinal nerve, not the dermatome (l 47). This should be corrected.

The authors repeat the statement that lumbar disc herniation is a commonly used term for LRP (l 50), which needs to be corrected. Lumbar disc herniation, in itself, does not necessarily cause symptoms.

The edited text “not favorable” (l 51) should be rephrased: Patients with LRP have not favorable prognosis in terms of pain and disability, as they need longer time to recover ….

The arguments made in the response letter regarding the choice of threshold of ≥ 60% for an acceptably would strengthen the overall quality of the paper and should be included in the Discussion section.

Misspelling: Additionally, it is important to include individuals form (from?) all relevant professions and take the view (l 328)

Author Response

We were delighted to read that this reviewer thinks our manuscript is substantially improved. Here we address his/her main concerns:

  • COMMENT: As I previously commented, radicular pain spreads along the spinal nerve, not the dermatome (l 47). This should be corrected. RESPONSE: we corrected this mistake (line 47).
  • COMMENT: The authors repeat the statement that lumbar disc herniation is a commonly used term for LRP (l 50), which needs to be corrected. Lumbar disc herniation, in itself, does not necessarily cause symptoms.  RESPONSE: We agree with the reviewer, therefore we adjusted the text accordingly (line 50).
  • COMMENT: The edited text “not favorable” (l 51) should be rephrased: Patients with LRP have not favorable prognosis in terms of pain and disability, as they need longer time to recover …. RESPONSE: We added the following text (l 51-55): "A large majority of patients with LRP tend to have a favorable prognosis in terms of pain and disability, but the time to recovery is usually longer than that of patients with LBP without concomitant radicular pain".
  • COMMENT: The arguments made in the response letter regarding the choice of threshold of ≥ 60% for an acceptably would strengthen the overall quality of the paper and should be included in the Discussion section. RESPONSE: Thank you for this suggestion, we now added the following text in the discussion (line 404-408): "We acknowledge the limitation of the AGREE II consortium not setting specific cut-off scores for domains or guidelines of high versus low quality. The cut-off scores used in this review were taken from a small study 26, but they were also used in other recent reviews of clinical practice guidelines using the AGREE II 61,62,63. More research on the most optimal cut-off scores for the domain and total scores is needed."
  • COMMENT: Misspelling: Additionally, it is important to include individuals form (from?) all relevant professions and take the view (l 328). RESPONSE: Amended (line 332).